# Antiviral Evaluation of UV-4B and Interferon-Alpha Combination Regimens against Dengue Virus

**DOI:** 10.3390/v13050771

**Published:** 2021-04-27

**Authors:** Evelyn J. Franco, Camilly P. Pires de Mello, Ashley N. Brown

**Affiliations:** 1Institute for Therapeutic Innovation, Department of Medicine, College of Medicine, University of Florida, Orlando, FL 32827, USA; e.franco@ufl.edu (E.J.F.); camillypestana@gmail.com (C.P.P.d.M.); 2Department of Pharmaceutics, College of Pharmacy, University of Florida, Orlando, FL 32827, USA

**Keywords:** DENV, antiviral, favipiravir, sofosbuvir, interferon, UV-4B

## Abstract

Dengue virus (DENV) is a flavivirus associated with clinical manifestations ranging in severity from self-limiting dengue fever, to the potentially life threatening condition, severe dengue. There are currently no approved antiviral therapies for the treatment of DENV. Here, we evaluated the antiviral potential of four broad-spectrum antivirals, UV-4B, interferon-alpha (IFN), sofosbuvir (SOF), and favipiravir (FAV) against DENV serotype 2 as mono- and combination therapy in cell lines that are physiologically relevant to human infection. Cell lines derived from human liver (HUH-7), neurons (SK-N-MC), and skin (HFF-1) were infected with DENV and treated with UV-4B, IFN, SOF, or FAV. Viral supernatant was sampled daily and infectious viral burden was quantified by plaque assay on Vero cells. Drug effect on cell proliferation in uninfected and infected cells was also assessed. UV-4B inhibited DENV in HUH-7, SK-N-MC, and HFF-1 cells yielding EC_50_ values of 23.75, 49.44, and 37.38 µM, respectively. Clinically achievable IFN concentrations substantially reduced viral burden in HUH-7 (EC_50_ = 102.7 IU/mL), SK-N-MC (EC_50_ = 86.59 IU/mL), and HFF-1 (EC_50_ = 163.1 IU/mL) cells. SOF potently inhibited DENV in HUH-7 cells but failed to produce the same effect in SK-N-MC and HFF-1 cells. Finally, FAV provided minimal suppression in HUH-7 and SK-N-MC cells, but was ineffective in HFF-1 cells. The two most potent anti-DENV agents, UV-4B and IFN, were also assessed in combination. UV-4B + IFN treatment enhanced antiviral activity in HUH-7, SK-N-MC, and HFF-1 cells relative to monotherapy. Our results demonstrate the antiviral potential of UV-4B and IFN against DENV in multiple physiologically relevant cell types.

## 1. Introduction

Dengue virus (DENV) is a mosquito-borne RNA virus belonging to the family *Flaviviridae* [1]. DENV is estimated to cause 400 million infections annually and is endemic to over one hundred countries, making it a major threat to global public health [2]. Infection is caused by one of four prevalent serotypes (DENV1, DENV2, DENV3 and DENV4) which commonly co-circulate in endemic areas [3]. The majority of cases result in asymptomatic infection (70–80% of cases) [4]; but, among symptomatic patients, clinical manifestations of infection can range in severity from self-limiting dengue fever to the more severe and potentially life threatening condition, severe dengue [3]. Risk of developing severe dengue increases with secondary infection with a serotype that is different from the primary DENV infection [1,4]. In addition to the classical spectrum of dengue clinical presentation, there is increasing evidence of neurologic complications associated with DENV including encephalitis, meningitis, and Guillain-Barré syndrome [5,6,7,8,9].

Despite DENV’s emergence as a global threat, there are currently no licensed antiviral treatments available. The availability and licensing of a tetravalent DENV vaccine (Dengvaxia^®^) has not replaced the need for effective antiviral therapies, as concerns over increased risk of severe dengue in seronegative individuals limits its use to patients with a confirmed prior DENV infection [10,11].

UV-4B, interferon-alpha (IFN), sofosbuvir (SOF), and favipiravir (FAV) are broad-spectrum antiviral agents that exhibit activity against multiple RNA viruses. UV-4B is an iminosugar antiviral that competitively inhibits host endoplasmic reticulum α-glucosidase I and II enzymes leading to misfolding of viral glycoproteins which is thought to decrease viral assembly/release or affect viral infectivity [12,13,14]. UV-4B has been previously investigated for its activity against RNA viruses such as DENV and influenza viruses [12,15,16]. IFN is a cytokine involved in activation of the host antiviral response [17]; prior to the approval of direct acting antivirals, its use in conjunction with ribavirin was the gold standard treatment for chronic hepatitis C virus (HCV) infection [18]. IFN has previously shown antiviral activity against DENV and other related flaviviruses [19,20,21,22]. SOF is a nucleotide polymerase inhibitor currently approved for treatment of chronic HCV infection and has also demonstrated antiviral activity against both ZIKV and DENV in vitro [23,24,25,26]. Finally, FAV is a nucleoside polymerase inhibitor that has exhibited a broad spectrum of activity against multiple RNA viruses. It is approved in Japan for the treatment of influenza virus and has been investigated in humans for its activity against Ebola virus during a 2014 outbreak in Guinea [27,28].

Previous anti-DENV assays with the above-mentioned agents were conducted using Vero cells, which are often considered to be the gold standard for viral research due to their high permissivity to viral infection. Vero cells, however, are derived from the kidneys of non-human primates and may not reflect the antiviral effect that would be achieved in man, as we have previously shown that viral susceptibility to antiviral therapy is highly variable based on the host cell line employed for the investigations [29,30]. Here, we evaluated the antiviral candidates against DENV in multiple cell lines derived from human tissue in an attempt to maximize translation to man. In addition to the selection of human cell lines, we also considered the fact that DENV exhibits a broad tissue tropism in vivo, meaning the virus can infect and replicate in multiple organs during human infection. Since DENV is pantropic, ideal antiviral candidates will inhibit viral replication in multiple target organs. Conducting antiviral evaluations in tissues that are representative of human target sites can lead to more accurate predictions of drug effect in afflicted organs. HUH-7 (human hepatocellular carcinoma) cells were selected because analysis of tissue samples from infected patients consistently show hepatocytes are key sites of viral replication [31,32]. SK-N-MC (human neuroepithelioma) cells were employed due to the risk for neurologic manifestations in cases of severe disease [31,33]. Finally, HFF-1 (human fibroblast) cells were utilized because skin cells are the first tissue type to be infected following viral transmission [31].

Due to their broad-spectrum antiviral activity, investigation of the potential anti-DENV activity of UV-4B, IFN, FAV and SOF is warranted. Therefore, our goal was to evaluate these compounds as monotherapy then select the most promising candidates as combination therapy against DENV in three clinically relevant human cell lines.

## 2. Materials and Methods

### 2.1. Cells, Virus and Compounds

Antiviral evaluations were carried out in three human cell lines. HUH-7 cells, were cultured in Dulbecco’s Modified Eagle’s Medium (DMEM; HyClone, Logan City, UT, USA) supplemented with 5% fetal bovine serum (FBS; Sigma-Aldrich, St. Louis, MO, USA) and 1% penicillin-streptomycin solution (HyClone). SK-N-MC (ATCC HTB-10) cells, were maintained in Minimum Essential Medium (MEM; Corning Cellgro, Corning, NY, USA) supplemented with 10% FBS, 1% penicillin-streptomycin solution, 1% sodium pyruvate (HyClone) and 1% non-essential amino acids solution (HyClone). HFF-1 (ATCC SCRC-1041) cells were maintained in DMEM supplemented with 15% FBS and 1% penicillin-streptomycin solution. Finally, our DENV plaque assays were performed on Vero (ATCC CCL-81) cells that were cultured in MEM supplemented with 5% FBS and 1% penicillin-streptomycin solution. All cell lines were incubated at 37 °C, 5% CO_2_ and subcultured twice weekly to maintain subconfluency.

Dengue virus serotype 2 (DENV2), strain New Guinea C was propagated on Vero cells and viral titers were determined as previously described [22]. Robust replication kinetics is vital for antiviral evaluations; therefore, DENV2 was chosen for these experiments, as this serotype demonstrated the most efficient replication kinetics in tissue culture relative to serotypes 1, 3, and 4.

UV-4B was kindly provided by Emergent BioSolutions Inc. (Gaithersburg, MD, USA) and IFN was obtained from PBL Assay Science (Piscataway, NJ, USA). FAV and SOF were purchased from MedKoo Biosciences, Inc. (Morrisville, NC, USA). All compounds were stored according to manufacturer recommendations. UV-4B stocks of 2809.8 µM (1000 µg/mL) were reconstituted in sterile deionized water. IFN stocks of 2 × 10^5^ IU/mL were prepared in PBS + 0.1% bovine serum albumin (BSA). FAV and SOF stocks of 50 mM and 10 mM, respectively, were reconstituted in 100% dimethyl sulfoxide (DMSO).

### 2.2. Antiviral Evaluations

HUH-7, SK-N-MC, and HFF-1 cells were infected with DENV at varying multiplicities of infection (MOIs) to maintain comparable viral replication kinetics between cell lines. A multiplicity of infection (MOI) of 0.01 was employed for HUH-7 cells, while SK-N-MC and HFF-1 cells were infected at an MOI of 0.1. Virus was allowed to adsorb onto cells for one hour, then the viral inoculum was removed. Monolayers were washed twice with warm PBS to remove unbound virus before drug containing medium was added to each well. For monotherapy evaluations, drug concentrations ranging from 0–400 µM UV-4B, 0–10,000 IU/mL IFN, 0–500 µM FAV, or 0–10 µM SOF were evaluated. Plates were maintained at 37 °C and 5% CO_2_. Viral supernatants were sampled daily over 3 days for HUH-7 cells, 4 days for SK-N-MC cells, and 2 days for HFF-1 cells, clarified by high speed centrifugation, and frozen at −80 °C until the end of the study. Viral burden in supernatant samples was quantified simultaneously via plaque assay on Vero cells as described previously [34]. Briefly, Vero cells were inoculated into 6-well plates and allowed to attach overnight at 37 °C, 5% CO_2_. The following day, viral supernatant samples were diluted serially 10-fold in MEM containing 2% FBS and each dilution was inoculated (0.1 mL) onto the confluent cell monolayers. Cells and virus were incubated for 1h at 37 °C, 5% CO_2_ and plates were shaken every 15 min to ensure dispersal of the viral inoculum over the cell monolayer. A primary agar overlay containing 0.6% agar, MEM, and 5% FBS was then added to cell monolayers and incubated at 37 °C, 5% CO_2_ for five days. A secondary overlay consisting of 1% agar, MEM, 1% FBS, 200 µg/mL diethylaminoethyl-dextran, and 0.008% neutral red solution was added and incubated overnight. Plaques were counted the next day. Assays were conducted in triplicate in two independent experiments to account for assay variability as well as between day variability.

For combination drug assays, HUH-7, SK-N-MC and HFF-1 cells were seeded into 6-well plates and infected as described above. All concentrations of UV-4B ranging from 0–400 µM and IFN 0–10,000 IU/mL were assessed either alone or in combination. Plates were maintained at 37 °C and 5% CO_2_. Viral supernatants were sampled on the day peak viral burden was achieved in the control (Day 1 for HFF-1 cells and Day 2 for HUH-7 and SK-N-MC cells), clarified by high-speed centrifugation, and frozen at −80°C. Viral burden was quantified via plaque assay on Vero cells [34]. Assays were conducted in triplicate in two independent experiments.

### 2.3. Cell Viability Assays

The commercially available WST-1 cell proliferation assay (Roche Diagnostics GmbH, Mannheim, Germany) was used to measure drug effect on cell proliferation and viability in uninfected and infected cells. HUH-7, SK-N-MC, or HFF-1 cells were seeded into 96-well plates at concentrations of 5000 cells/well for HUH-7 and HFF-1 cells and 25,000 cells/well for SK-N-MC cells and incubated overnight at 37 °C in 5% CO_2_. The following day, medium in each well was removed and cells were either mock infected with viral diluent (MEM + 2% FBS), or infected with DENV at MOIs of 0.01 for HUH-7 cells, 0.1 for SK-N-MC cells, and 1 for HFF-1 cells. Drug containing medium at the same concentrations used during monotherapy assays was added to each well 4 h after inoculation with virus or diluent. Infected and uninfected cells were incubated in the presence of drug for 3 days, then WST-1 reagent was added to each well as per manufacturer recommendations. Absorbance was detected 2–4 h after reagent was added using a SpectraMax M5 microplate reader (Molecular Devices, San Jose, CA, USA).

### 2.4. Statistical Analysis

To determine the EC_50_ values for monotherapy studies, an inhibitory sigmoid-E_max_ model was fit to the area under the viral burden-time curve (AUC_viral_burden_) over the entire time course of the experiment using GraphPad Prism software (GraphPad Software; La Jolla, CA, USA). CC_50_ values were calculated by graphing absorbance values at 72 h against drug concentration and fitting an inhibitory sigmoid E_max_ model to the data using GraphPad Prism software. Cell viability in infected cells is reported as percent cell viability relative to an untreated, uninfected control.

## 3. Results

### 3.1. UV-4B

In HUH-7 cells, UV-4B yielded an EC_50_ value of 23.75 µM (Table 1). Lower drug concentrations did not effectively suppress DENV as viral titers were nearly identical to the control arm, but noticeable viral suppression was achieved at concentrations ≥ 25 µM. Antiviral effect was lost by day 3 in all treatment arms suggesting drug treatment delayed, but did not fully inhibit, production of infectious virus in this cell line (Figure 1A). UV-4B exhibited an EC_50_ of 49.44 µM (Table 1, Appendix A) in SK-N-MC cells. Treatment inhibited DENV (Figure 1B) in a concentration dependent manner in this cell line; on day 2, effective regimens (25 µM–400 µM) decreased titers by 1.6- and 3.6-log10 PFU/mL at 25 and 400 µM, respectively. Similar to HUH-7 cells, viral inhibition in HFF-1 cells was most pronounced at UV-4B concentrations ≥ 25 µM. Antiviral effect was maintained throughout the course of therapy in this cell line (Figure 1C), and UV-4B achieved an EC_50_ value equivalent to 37.38 µM.

Each drug was evaluated for potential cytotoxic effects through cell proliferation assays in uninfected cells. Our results indicated HUH-7 cells were most susceptible to UV-4B cytotoxicity relative to SK-N-MC and HFF-1 cells. High concentrations of UV-4B (≥100 µM) were especially toxic to HUH-7 cells, yielding a CC_50_ of 135 µM (Table 1). In contrast, evaluated concentrations of UV-4B caused no detectable cytotoxicity in SK-N-MC cells (CC_50_ > 400 µM) (Table 1). HFF-1 cells were susceptible to UV-4B toxicity but decline in cell viability was slight, indicating exposure to high concentrations of drug leads to a cytostatic effect in this cell line (CC_50_ > 400 µM) (Table 1). Cell proliferation assays were also employed in infected cells to determine the extent to which treatment protected cells from death or damage as a result of infection. Results in HUH-7 cells showed infection did not cause substantial cell death in this experiment since percent cell viability in untreated, infected cells was not much lower than an uninfected control. Cell viability measurements remained comparable to the infected control following exposure to UV-4B at concentrations of up to 25 µM. The declines in cell viability observed at UV-4B concentrations of 100 and 400 µM are due to HUH-7 cell sensitivity to UV-4B toxicity (Figure 1D). In the absence of treatment and at low UV-4B concentrations, infection causes extensive cytopathic effect (CPE) and cell death in SK-N-MC cells, but treatment with UV-4B at concentrations ≥ 25 µM caused dramatic increases in cell viability indicating these concentrations of drug improve cell health and protect infected cells from formation of CPE and death (Figure 1E). DENV infection does not cause CPE in HFF-1 cells, thus cell viability of infected treatment arms exposed to drug concentrations ranging from 0–25 µM does not greatly differ from the uninfected control. The moderate declines in viability observed at 100 and 400 µM reflect UV-4B cytostatic effect on this cell line at high drug concentrations (Figure 1F).

### 3.2. IFN

IFN caused robust, sustained viral inhibition in HUH-7 cells, exhibiting an EC_50_ value of 102.7 IU/mL (Table 1). The clinically achievable concentration of 100 IU/mL reduced peak titers by 2.4 log10 PFU/mL, while the highest concentration evaluated (10,000 IU/mL) completely curbed the production of infectious virus (Figure 2A). IFN inhibited DENV in a concentration dependent manner in SK-N-MC cells, yielding an EC_50_ of 86.59 IU/mL. Antiviral activity was transient in this cell line however, as viral burden in all treatment arms continuously increased throughout the experiment (Figure 2B). Finally, a dose-response effect was observed when infected HFF-1 cells were treated with IFN. 100 IU/mL IFN yielded a 1.6 log10 PFU/mL decrease in viral burden relative to the no treatment control (Figure 2C). The EC_50_ in this cell line was 163.1 IU/mL

Cytotoxicity studies in uninfected cells demonstrated that IFN caused no cytotoxic effect in HUH-7 and HFF-1 cells at evaluated concentrations (CC_50_s > 10,000 IU/mL) (Table 1, Appendix A). IFN concentrations ranging from 1–1000 IU/mL were similarly nontoxic to SK-N-MC cells, but cytotoxicity was detected at drug concentrations greater than 1000 IU/mL, yielding a CC_50_ value of 1074 IU/mL (Table 1, Appendix A). Assays conducted in infected cells showed IFN protects HUH-7 and SK-N-MC cells from cell damage and death associated with infection. This protective effect was most pronounced in HUH-7 cells, where treatment with IFN at concentrations ≥ 10 IU/mL markedly increased cell viability relative to untreated, infected cells (Figure 2D). Exposure to nontoxic concentrations of IFN also increased viability of infected SK-N-MC cells in a concentration dependent manner (Figure 2E), but this cell line was not as sensitive to IFN effect so improvements in percent cell viability were less than those observed in HUH-7 cells. Cell viability of infected HFF-1 cells was not appreciably different from the uninfected control because infection does not cause CPE or cell death in this cell line (Figure 2F). Percent cell viability stayed constant following addition of increasing IFN concentrations, further illustrating this drug’s lack of cytotoxic effects at evaluated concentrations.

### 3.3. SOF

SOF caused effective, sustained viral suppression at concentrations ≥4 µM in HUH-7 cells (EC_50_ = 6.96 µM) (Table 1). Treatment with 4 and 6 µM SOF resulted in a nearly 100-fold reduction in viral burden and the highest concentration evaluated (10 µM) drove viral titers below the limit of detection (Figure 3A). In contrast, neither SK-N-MC nor HFF-1 cells were susceptible to SOF antiviral effect, as viral burden in all treatment arms remained comparable to the no treatment control in both cell lines (Figure 3B,C). Cell viability studies conducted in uninfected cells showed SOF caused no cytotoxicity in all three cell lines (Table 1, Appendix A). In addition to decreasing DENV titers, exposing infected HUH-7 cells to SOF also yielded improvements in cell viability; this effect was especially apparent at drug concentrations of 4 µM and greater (Figure 3D). Cell proliferation studies conducted in infected SK-N-MC cells showed SOF exposure caused minimal changes in cell viability (Figure 3E). As was seen with the other drugs, cell viability of infected HFF-1 cells did not change relative to the uninfected control, indicating that neither infection nor exposure to increasing concentrations of SOF led to extensive cell death (Figure 3F).

### 3.4. FAV

Viral inhibition by FAV was slight in HUH-7 cells (EC_50_ = 146.8 µM). 250 µM FAV resulted in an approximate tenfold decrease in titers relative to the control; exposure to higher drug concentrations did not substantially enhance viral suppression since a maximum 1.2 log10 PFU/mL reduction was achieved by the highest concentration evaluated (Figure 4A). FAV’s antiviral activity was also minimal in SK-N-MC cells (EC_50_ = 287.9 µM). 250 µM FAV lowered viral titers by approximately 0.6 log10 PFU/mL on Day 2 of therapy and 500 µM yielded an additional 0.7 log10 PFU/mL (total of 1.3 log10 PFU/mL reduction) decrease (Figure 4B). FAV was completely ineffective against DENV in HFF-1 cells as viral burden in all treatment arms was identical to the no treatment control (Figure 4C). FAV caused no cytotoxicity at evaluated concentrations (Table 1, Appendix A). Cell proliferation assays in infected cells showed FAV exposure had limited effect on cell health in all three cell lines (Figure 4D–F).

### 3.5. Evaluation of UV-4B and IFN as Combination Therapy

Six concentrations of IFN (0–10,000 IU/mL), and UV-4B (0–400 µM) were evaluated alone and in every possible combination of concentrations against DENV in HUH-7, SK-N-MC, and HFF-1 cells. DENV achieved a peak viral burden of 7 log10 PFU/mL on day 2 in HUH-7 cells. UV-4B and IFN combination regimens augmented antiviral activity relative to monotherapy in this cell line; this effect became particularly apparent when UV-4B concentrations ≥ 25 µM were added to IFN. IFN 100 IU/mL + UV-4B 25 µM reduced viral burden relative to the no treatment control by 3.5 log10 PFU/mL while monotherapy at these concentrations lowered viral titers by 2.5 and 1.7 log10 PFU/mL, respectively (Figure 5A).

In the absence of therapy, DENV titers peaked at 6.2 log10 PFU/mL in SK-N-MC cells. UV-4B caused considerable viral suppression as monotherapy, yielding a decline in peak viral titers of 3.8 log10 PFU/mL at the highest concentration evaluated (400 µM). IFN monotherapy also curbed production of infectious virus; the clinically achievable concentration of 100 IU/mL, and the highest concentration evaluated, 10,000 IU/mL, lowered peak viral titers by 1.4 and 2.5 log10 PFU/mL, respectively. Combination regimens slightly enhanced viral inhibition in this cell line. Treatment with 100 IU/mL plus 25 µM UV-4B reduced peak viral titers by 2.2 log10 PFU/mL relative to the no treatment control. This combination suppressed DENV replication by an additional 0.8 log10 PFU/mL relative to 100 IU/mL IFN as monotherapy and 0.6 log10 PFU/mL compared to 25 µM UV-4B as monotherapy (Figure 5B).

In HFF-1 cells, DENV viral burden peaked at 4.3 log10 PFU/mL on day 1. The therapeutically achievable IFN concentration of 100 IU/mL effectively inhibited viral replication, and addition of UV-4B at all evaluated concentrations drove viral burden to below the limit of detection (Figure 5C). Addition of UV-4B to IFN 10 IU/mL enhanced IFN’s antiviral effect, particularly at concentrations ≥ 6.25 µM. IFN 10 IU/mL + 25 µM UV-4B caused a 2.3 log10 PFU/mL reduction in viral titers, nearly an additional ten-fold unit of viral suppression relative to IFN monotherapy at 10 IU/mL and 1.3 log10 PFU/mL relative to UV-4B alone at 25 µM.

## 4. Discussion

DENV is responsible for 100–400 million infections per year [3]. Despite the substantial burden imposed by DENV on global public health, treatment recommendations remain nonspecific, with no antivirals approved to prevent infection or alleviate symptoms in infected patients. Here, we aimed to evaluate the antiviral potential of four broad spectrum antivirals. We have previously shown antiviral effectiveness can be heavily influenced by cell line [29], hence each compound has been evaluated for its anti-DENV activity in a variety of human cell lines that are physiologically relevant to DENV infection.

UV-4B inhibited DENV in all three cell lines evaluated. These results are in accordance with work done by others who have shown dengue virus is susceptible to UV-4B and related α-glucosidase inhibitors both in vitro and in vivo [12,14,15]. In addition to inhibiting production of infectious DENV, UV-4B treatment caused a concentration dependent increase in cell viability in SK-N-MC cells. Viral inhibition is thought to occur either through (1) reduced viral assembly and release or (2) altering of viral infectivity [12,13]. Both of these pathways reduce the amount of replication competent virus available to infect cells which plausibly explains the reductions in viral titers and increases in cell viability produced when infected SK-N-MC cells were treated with UV-4B. UV-4B inhibited DENV in HUH-7 and HFF-1 cells however, both cell lines were susceptible to UV-4B toxicity. Drug concentrations ≥ 100 µM caused a dramatic decrease in cell viability in HUH-7 cells and a more modest decline in HFF-1 cells. This cytotoxic effect in HUH-7 cells and cytostatic effect in HFF-1 cells likely contributed to apparent UV-4B activity at high concentrations as viral replication is dependent on the presence of actively replicating cells. UV-4B has been investigated as a potential therapeutic option against DENV and influenza viruses [15,16]; however, its broad spectrum activity is likely to extend to other flaviviruses as these are also dependent on ER α-glucosidases for proper processing and folding of envelope glycoproteins [35,36,37]. This is an active are of investigation in our laboratory, as UV-4B holds promise as an antiviral candidate with broad spectrum activity against other flaviviruses including Zika and West Nile virus.

We have previously shown that IFN holds potential as an antiviral candidate against DENV [22]. Our results showed IFN inhibited DENV replication in all cell lines yielding EC_50_ values of 102.7, 86.59, and 163.1 IU/mL in HUH-7, SK-N-MC, and HFF-1 cells, respectively. At steady state, clinical IFN regimens reach average drug concentrations equivalent to approximately 200 IU/mL [38]. Because the calculated EC_50_ values from our studies fell within IFN’s therapeutic range, further study of IFN’s promise as a treatment strategy against DENV is warranted. In addition to curbing production of infectious virus, further evidence of IFN’s protective effect was seen in our cell proliferation assays in infected cells. Here, the presence of increasing concentrations of IFN improved cell viability in infected HUH-7 cells at all evaluated concentrations and in SK-N-MC cells at concentrations of 1–1000 IU/mL (decline in cell viability at 10,000 IU/mL due to cytotoxic effects at concentrations >1000 IU/mL). Binding of IFN to cell surface receptors leads to establishment of an antiviral state where the production of antiviral genes and proteins is upregulated within the cell [18,39], these cellular changes disrupt the viral replication cycle and hinder cell to cell virus spread [17]. IFN treatment likely limited the amount of cells that were susceptible to infection following exposure to DENV thus explaining the increases in cell viability observed in HUH-7 and SK-N-MC cells.

Although therapeutically feasible concentrations of IFN suppressed DENV in all three cell lines, SK-N-MC cells are less susceptible to IFN effect relative to HUH-7 and HFF-1 cells. For example, 100 IU/mL IFN inhibited DENV by 2.4 log10 PFU/mL in HUH-7 cells while 10,000 IU/mL were required to elicit a similar degree of suppression in SK-N-MC cells. Additionally, the antiviral activity exerted by 10,000 IU/mL IFN in HFF-1 cells was tenfold higher than that achieved by the same concentration of drug in SK-N-MC cells. This may be due to differences in IFN receptor expression in SK-N-MC cells relative to HUH-7 and HFF-1 cells. Others have shown IFN potency is related to the abundance of cell surface receptors and that greater IFN sensitivity could be achieved through higher expression of IFN receptors [40]. It is plausible SK-N-MC cells express fewer IFN receptors on their surface which could contribute to reduced sensitivity to IFN effect.

SOF caused substantial viral inhibition in HUH-7 cells but was ultimately ineffective in SK-N-MC and HFF-1 cells. SOF’s robust activity in HUH-7 cells is not unexpected for several reasons. First, SOF was specifically developed for the treatment of chronic hepatitis C infection; thus, it was designed to yield substantial uptake and metabolism in liver cells [23,41]. Second, our results are in agreement with those of others who have similarly shown its potent anti-DENV effect in hepatocytes [24,42]. Cell line dependent variability in the same factors favoring sofosbuvir activation in HUH-7 cells may limit its effectiveness in other tissues thereby explaining lack of activity in SK-N-MC and HFF-1 cells. For example, the drug’s ability to penetrate cell membranes may be host cell dependent. Alternatively, tissue specific differences in intracellular concentrations of enzymes such as esterases and kinases may prevent unmasking of the molecule’s monophosphate moiety or affect formation and accumulation of the active triphosphate metabolite [41,42,43]. For these studies, we sought to identify broad spectrum antivirals that inhibited production of infectious DENV in multiple cell types; since only hepatocytes were sensitive to SOF effect, it was not selected for further study.

Our results suggest FAV is not a feasible candidate for future study since clinically feasible drug concentrations caused only slight inhibition in HUH-7 and SK-N-MC cells and were completely ineffective in HFF-1 cells. Since FAV is administered as a prodrug whose effectiveness is linked to efficient drug uptake and activation by the host cell, it is possible tissue dependent variability in these processes impacted drug effect. These considerations explain the absence of activity in HFF-1 cells, as work done in our laboratory evaluating intracellular concentrations of FAV and FAV-RTP demonstrate FAV does not effectively penetrate HFF-1 cell membranes [30], but they do not adequately explain lack of effect in HUH-7 and SK-N-MC cells. Studies investigating FAV’s antiviral potential against Zika virus (ZIKV) conducted in our lab have shown robust inhibition of ZIKV in both cell lines [21] indicating FAV uptake and activation is efficient in these cells. It is possible DENV’s insensitivity to FAV in these studies is because FAV is not an effective substrate of DENV’s RNA polymerase. In order to effect RNA replication, the nucleoside analogue must be recognized by the viral polymerase as a potential substrate and compete with endogenous nucleotides for incorporation to the nascent RNA strand [44]. Further study would be required to demonstrate whether DENV’s RNA polymerase may have a higher affinity for endogenous purines over FAV, leading to infrequent incorporation of the analogue and little antiviral effect.

Due to their demonstrated effect against DENV in a range of cell types, UV-4B and IFN were selected for further study as combination regimens. Our in vitro evaluations demonstrated combination therapy with IFN plus UV-4B enhanced antiviral effect in HUH-7, SK-N-MC, and HFF-1 cells relative to monotherapy. These results are encouraging for several reasons. First, because any further reduction in viral titers would be beneficial in keeping DENV replication in check until the immune system mounts a robust response to eliminate the virus. Second, because combination regimens enhanced viral suppression at therapeutically feasible concentrations of both agents, since clinical regimens of IFN reach exposures of approximately 200 IU/mL [38], while a Phase I study showed a one-time dose of 1000 mg UV-4B was associated with a C_max_ of approximately 40 µM [45]. The enhanced effect of combinations of UV-4B plus IFN at therapeutically feasible concentrations was most pronounced in HUH-7 and HFF-1 cells because of their strong susceptibility to both antivirals. Additional viral suppression from combination regimens was subtler in SK-N-MC cells, due to their lower susceptibility to IFN. Since IFN is associated with an unfavorable side effect profile, combining IFN with UV-4B could spare the amount of IFN required to treat infection.

Traditionally, regimens combining antivirals are designed with 1 of 2 goals in mind, (1) preventing emergence of viral resistance, or (2) producing additive or synergistic drug effects. Here, we present an alternative way to approach antiviral combination regimens; by combining agents with broad spectrum activity in a range of target tissues, we can disrupt viral replication and spread in multiple affected organs which will allow for better treatment of pantropic viruses such as DENV that establish infection in multiple tissue types.

There were several limitations to this study. First, all treatment regimens were evaluated at static drug concentrations, this does not accurately represent the pharmacokinetic profiles achieved following drug administration in man. Future studies using the hollow fiber infection model will allow us to simulate the PK profiles associated with UV-4B and IFN administration in man and provide insight into how these agents would behave under dynamic concentrations. Second, these regimens were evaluated against a serotype 2 strain of DENV, future studies will be conducted to evaluate the effectiveness of these regimens against the other three commonly circulating DENV serotypes.

The results of this study demonstrate the antiviral potential of UV-4B and IFN combination regimens against DENV. Treatment strategies against pantropic viruses such as DENV would ideally inhibit viral replication in all affected tissues, here we have presented a potential approach to achieve this goal. By combining antivirals with demonstrated activity in various tissue types, we can expand the number of affected organs that are treated by one or both agents in the combination.

## Figures and Tables

**Figure 1 viruses-13-00771-f001:**
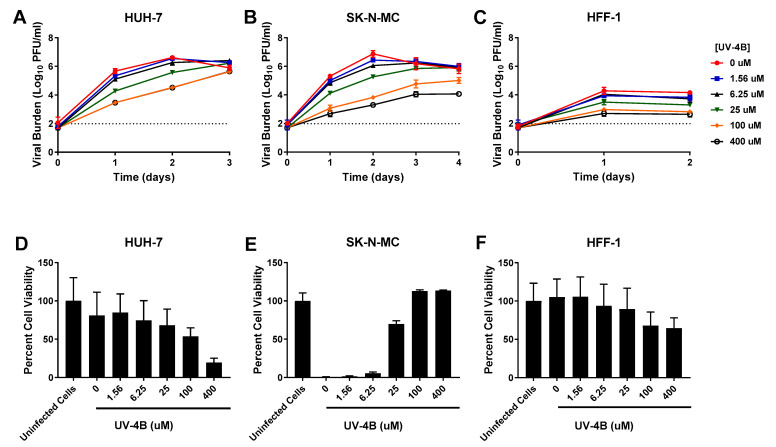
Antiviral activity of UV-4B against dengue virus (DENV) in HUH-7, SK-N-MC, and HFF-1 cells (**A**–**C**). HUH-7 (**A**) cells were infected at a multiplicity of infection (MOI) of 0.01, and SK-N-MC (**B**) and HFF-1 (**C**) cells were infected at an MOI of 0.1. Infected cells were treated with different concentrations of UV-4B. Viral burden was quantified via plaque assay on Vero cells, and reported as log10 plaque forming units per mL (PFU/mL). Data points represent the mean of three independent samples, and error bars correspond to one standard deviation. The dashed line signifies the assay limit of detection. (**D**–**F**) Effect of UV-4B treatment on cell viability of DENV-infected HUH-7 (**D**), SK-N-MC (**E**), and HFF-1 (**F**) cells. HUH-7, SK-N-MC, and HFF-1 cells were infected with DENV at MOIs of 0.01, 0.1, and 1, respectively, then treated with different concentrations of UV-4B. Cell viability was measured after three days with the commercially available WST-1 assay according to manufacturer specifications. Cell viability is reported as percent cell viability relative to an untreated, uninfected control. Columns represent the mean of 6 independent samples, error bars represent one standard deviation.

**Figure 2 viruses-13-00771-f002:**
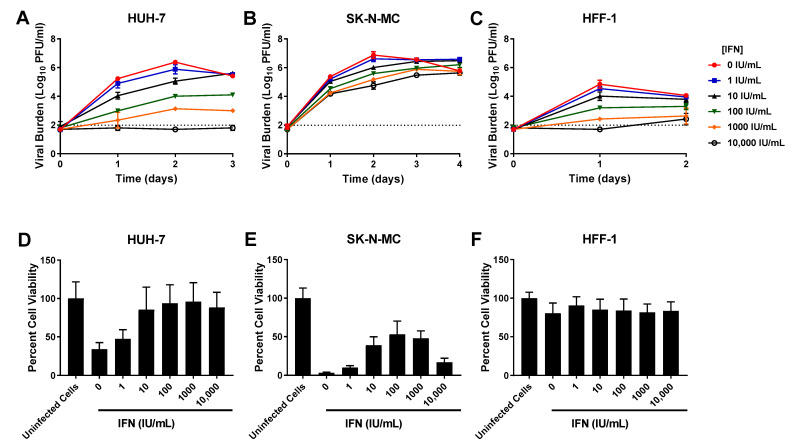
Antiviral activity of interferon-alpha (IFN) against dengue virus (DENV) in HUH-7, SK-N-MC, and HFF-1 cells (**A**–**C**). HUH-7 (**A**) cells were infected at a multiplicity of infection (MOI) of 0.01, and SK-N-MC (**B**) and HFF-1 (**C**) cells were infected at an MOI of 0.1. Infected cells were treated with different concentrations of IFN. Viral burden was quantified via plaque assay on Vero cells, and reported as log10 plaque forming units per mL (PFU/mL). Data points represent the mean of three independent samples, and error bars correspond to one standard deviation. The dashed line signifies the assay limit of detection. (**D**–**F**) Effect of IFN treatment on cell viability of DENV-infected HUH-7 (**D**), SK-N-MC (**E**), and HFF-1 (**F**) cells. HUH-7, SK-N-MC, and HFF-1 cells were infected with DENV at MOIs of 0.01, 0.1, and 1, respectively, then treated with different concentrations of IFN. Cell viability was measured after three days with the commercially available WST-1 assay according to manufacturer specifications. Cell viability is reported as percent cell viability relative to an untreated, uninfected control. Columns represent the mean of 6 independent samples, error bars represent one standard deviation.

**Figure 3 viruses-13-00771-f003:**
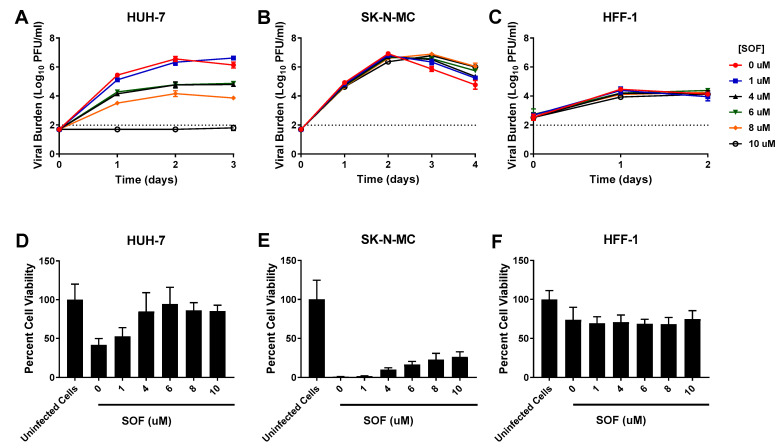
Antiviral activity of sofosbuvir (SOF) against dengue virus (DENV) in HUH-7, SK-N-MC, and HFF-1 cells (**A**–**C**). HUH-7 (**A**) cells were infected at a multiplicity of infection (MOI) of 0.01, and SK-N-MC (**B**) and HFF-1 (**C**) cells were infected at an MOI of 0.1. Infected cells were treated with different concentrations of SOF. Viral burden was quantified via plaque assay on Vero cells, and reported as log10 plaque forming units per mL (PFU/mL). Data points represent the mean of three independent samples, and error bars correspond to one standard deviation. The dashed line signifies the assay limit of detection. (**D**–**F**) Effect of SOF treatment on cell viability of DENV-infected HUH-7 (**D**), SK-N-MC (**E**), and HFF-1 (**F**) cells. HUH-7, SK-N-MC, and HFF-1 cells were infected with DENV at MOIs of 0.01, 0.1, and 1, respectively, then treated with different concentrations of SOF. Cell viability was measured after three days with the commercially available WST-1 assay according to manufacturer specifications. Cell viability is reported as percent cell viability relative to an untreated, uninfected control. Columns represent the mean of 6 independent samples, error bars represent one standard deviation.

**Figure 4 viruses-13-00771-f004:**
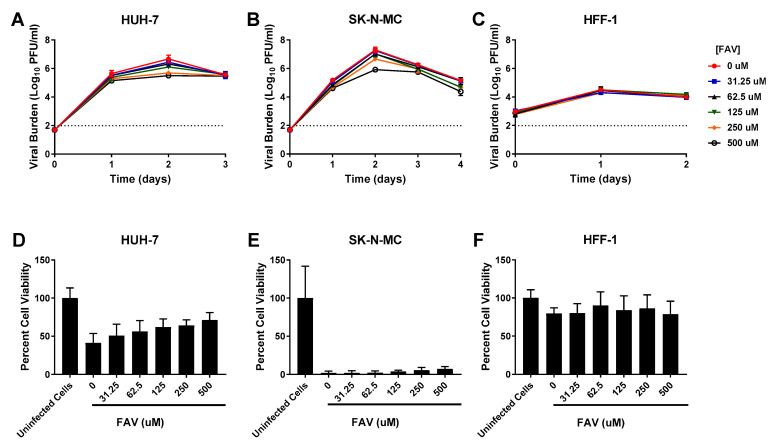
Antiviral activity of favipiravir (FAV) against dengue virus (DENV) in HUH-7, SK-N-MC, and HFF-1 cells (**A**–**C**). HUH-7 (**A**) cells were infected at a multiplicity of infection (MOI) of 0.01, and SK-N-MC (**B**) and HFF-1 (**C**) cells were infected at an MOI of 0.1. Infected cells were treated with different concentrations of FAV. Viral burden was quantified via plaque assay on Vero cells, and reported as log10 plaque forming units per mL (PFU/mL). Data points represent the mean of three independent samples, and error bars correspond to one standard deviation. The dashed line signifies the assay limit of detection. (**D**–**F**) Effect of FAV treatment on cell viability of DENV-infected HUH-7 (**D**), SK-N-MC (**E**), and HFF-1 (**F**) cells. HUH-7, SK-N-MC, and HFF-1 cells were infected with DENV at MOIs of 0.01, 0.1, and 1, respectively, then treated with different concentrations of FAV. Cell viability was measured after three days with the commercially available WST-1 assay according to manufacturer specifications. Cell viability is reported as percent cell viability relative to an untreated, uninfected control. Columns represent the mean of 6 independent samples, error bars represent one standard deviation.

**Figure 5 viruses-13-00771-f005:**
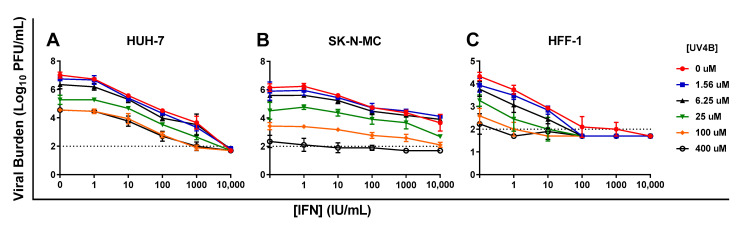
Antiviral activity of UV-4B and interferon-alpha (IFN) combination therapy against dengue virus (DENV) in HUH-7 (**A**), SK-N-MC (**B**), and HFF-1 (**C**) cells. HUH-7 cells were infected with DENV at an MOI of 0.01, and SK-N-MC and HFF-1 cells were infected at an MOI of 0.1 Different concentrations of UV-4B, IFN, or both were added to cells following infection. Viral supernatant was sampled when peak viral titers were achieved in the control (Day 1 for HFF-1 cells and Day 2 for HUH-7 and SK-N-MC cells) and viral burden was quantified via plaque assay on Vero cells. Data points represent the mean of three independent samples and error bars correspond to one standard deviation. The dashed line signifies the assay limit of detection.

**Table 1 viruses-13-00771-t001:** Fifty percent effective concentration (EC_50_) and fifty percent cytotoxic concentration (CC_50_) of UV-4B, Interferon-alpha (IFN), Sofosbuvir (SOF), and favipiravir (FAV) in HUH-7, SK-N-MC, and HFF-1 cells.

	HUH-7 Cells		SK-N-MC Cells		HFF-1 Cells	
Drug	EC_50_	CC_50_	EC_50_	CC_50_	EC_50_	CC_50_
UV-4B (µM)	23.75	135	49.44	>400	37.38	>400
IFN (IU/mL)	102.7	>10,000	86.59	1074	163.1	>10,000
SOF (µM)	6.96	>10	>10	>10	>10	>10
FAV (µM)	148.8	>500	287.9	>500	>500	>500

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
