# Peer review of "Antiviral Evaluation of UV-4B and Interferon-Alpha Combination Regimens against Dengue Virus"

_viruses, 2021, doi:10.3390/v13050771_

Round 1
Reviewer 1 Report
Manuscript entitled "Antiviral Evaluation of UV-4B and Interferon-Alpha Combination Regimens against Dengue Virus" by authors Evelyn J. Franco et al is well designed and the results are presented well.
Authors have made an attempt to study the effect of UV-48, IFN-alpha, SOF and FAV against Dengue virus infection in skin cells, brain cells and hepatic cells. Authors have determined the cytotoxicity effect of these therapeutic agents at various concentrations and in combination of IFN & UV-48.
Authors have mentioned that they propagated the virus on Vero cells. Did authors try to propagate on C6/36 (mosquito cell line) also?
It would be good, if authors can briefly explain the plaque assay procedure and present some of the representative images in the supplemental (this will help other researchers to refer)
Authors may need to modify the figures clearly (X axis and Y axis font size needs to be changed or make it bold for better visibility)
Authors have also mentioned about the limitations of this study, so that other researchers can be careful about designing such study.
Authors have discussed the results in detail and overall the manuscript is good.
Author Response
Manuscript entitled "Antiviral Evaluation of UV-4B and Interferon-Alpha Combination Regimens against Dengue Virus" by authors Evelyn J. Franco et al is well designed and the results are presented well.
Authors have made an attempt to study the effect of UV-48, IFN-alpha, SOF and FAV against Dengue virus infection in skin cells, brain cells and hepatic cells. Authors have determined the cytotoxicity effect of these therapeutic agents at various concentrations and in combination of IFN & UV-48.
Authors have mentioned that they propagated the virus on Vero cells. Did authors try to propagate on C6/36 (mosquito cell line) also?
Response: Thank you for your kind comments regarding our manuscript. To answer this specific question, yes, we have propagated Dengue virus in C6/36 cells. However, viral replication kinetics in this cell line were substantially slower and not as robust when compared to Vero cells. This resulted in viral stocks that had a markedly lower viral titer. We performed some antiviral experiments with a dengue virus stock that was propagated in C6/36 cells and the results were very similar to those observed here using Vero derived stock virus. Thus, we chose to perform our larger studies (evaluating multiple drugs on multiple cells lines) using Vero cell stocks as the viral burden in these stocks were higher, resulting in the need of less virus. Vero cell stocks are often used for experimentation with Dengue virus as demonstrated in the literature, so this protocol is not out of the ordinary.
It would be good, if authors can briefly explain the plaque assay procedure and present some of the representative images in the supplemental (this will help other researchers to refer)
Response: A brief explanation highlighting the methods of the plaque assay have been added to the manuscript in lines 126-135. The viral plaque assay is a standard method used to quantify the concentration of infectious virus in a sample and is ubiquitously used in the field. Our plaque assay plates look exactly like those reported by others using a neutral red-based plaque assay. We obtained clear plaques that were relatively homogeneous in size (our results look exactly like the images presented when the phrase neutral red plaque assay is googled). Due to the ordinary nature of our plaque assay, we chose to not include representative images of our results.
Authors may need to modify the figures clearly (X axis and Y axis font size needs to be changed or make it bold for better visibility)
Response: Thank you for the suggestion. The change has been made so that the fonts are larger and bolded for ease of reading.
Authors have also mentioned about the limitations of this study, so that other researchers can be careful about designing such study.
Authors have discussed the results in detail and overall the manuscript is good.
Response: Again, thank you for the kind words regarding our manuscript.
Reviewer 2 Report
In this manuscript, the authors present data demonstrating antiviral effects of 4 compounds: UV-4B, IFN-a, sofusbivir, and favipiravir, against DENV2 in a range of cell types with relevance to flavivirus infection (Huh7—hepatic, SK-N-MC—neuronal, HFF1—fibroblasts).
Figure 1-4 (D-F): Rather than showing the effect on cell viability of each compound in the context of infection, it would be better to show toxicity data for uninfected cells. Although this data is provided in Table 1, it should be presented in graph form as well. It’s too difficult to parse out the cytopathic effect of viral infection from cytotoxic effect in the data as the data is currently presented.
Figure 5: It is difficult to recognize from this figure the value of combination therapy because there is so much data packed into each graph. Perhaps it would be useful to include an isobologram or combination index analysis—similar to (Feng, J.Y., Ly, J.K., Myrick, F. et al. The triple combination of tenofovir, emtricitabine and efavirenz shows synergistic anti-HIV-1 activity in vitro: a mechanism of action study. Retrovirology 6, 44 (2009). https://doi.org/10.1186/1742-4690-6-44) for example.
Alternatively, since as you indicate in your discussion, IFN treatment is associated with considerable negative in vivo side effects, you could choose an ideal IFN concentration and represent the benefit of increasing concentrations of UV-4B in that context.
Author Response
In this manuscript, the authors present data demonstrating antiviral effects of 4 compounds: UV-4B, IFN-a, sofusbivir, and favipiravir, against DENV2 in a range of cell types with relevance to flavivirus infection (Huh7—hepatic, SK-N-MC—neuronal, HFF1—fibroblasts).
Figure 1-4 (D-F): Rather than showing the effect on cell viability of each compound in the context of infection, it would be better to show toxicity data for uninfected cells. Although this data is provided in Table 1, it should be presented in graph form as well. It’s too difficult to parse out the cytopathic effect of viral infection from cytotoxic effect in the data as the data is currently presented.
Response: We agree with the reviewer that cytotoxicity profiles of drugs are easier to view graphically. However, we also believe that it is beneficial to show the antiviral activity of these drugs using different experimental endpoints, such as the cell protection assays as well as the viral burden assays shown in our manuscript. Thus, we have included the cytotoxicity profiles for each drug on uninfected HUH-7, SK-N-MC, and HFF-1 cells to demonstrate cytotoxicity associated with treatment as a supplementary figure. Due to the size of this figure (12 panels), we thought it was best to include it as supplemental material.
Figure 5: It is difficult to recognize from this figure the value of combination therapy because there is so much data packed into each graph. Perhaps it would be useful to include an isobologram or combination index analysis—similar to (Feng, J.Y., Ly, J.K., Myrick, F. et al. The triple combination of tenofovir, emtricitabine and efavirenz shows synergistic anti-HIV-1 activity in vitro: a mechanism of action study. Retrovirology 6, 44 (2009). https://doi.org/10.1186/1742-4690-6-44) for example. Alternatively, since as you indicate in your discussion, IFN treatment is associated with considerable negative in vivo side effects, you could choose an ideal IFN concentration and represent the benefit of increasing concentrations of UV-4B in that context.
Response: We agree with the reviewer that the combination results do present a lot of data. However, we find that the way the data is presented in the manuscript is the easiest form to view this information. The isobologram or combination index analysis suggested by the reviewer is an excellent idea, but requires mathematical analyses that are outside the scope of this manuscript. As discussed in our manuscript, we are viewing the benefits of combination therapy in a way that lies outside the “normal” use. While we would like to see enhanced antiviral activity (synergy or additivity), it is not a criteria for success. As long as we do not observe antagonism in which the addition of one drug actually prevents action by the second drug, we consider the combination a success. We are looking for at least one drug in the combination to be effective in a given cell type. We think this may be a strategy to target viral replication in multiple tissues that have varying susceptibilities to drug treatment.
Graphically, the way the data is presented, enhanced antiviral activity is demonstrated by lower viral burden in a combination arm compared to monotherapy. In figure 5, UV-4B monotherapy is illustrated in red and IFN monotherapy is shown by the points on the y-axis. For example, in HUH-7 cells, the combination of 100 IU/ml of IFN with 100 uM of UV-4B results in a lower viral burden compared to either agent as monotherapy. This is explained in the results section of the manuscript. We feel as though 3D plots of the data are more difficult to read, which is why we often select this type of graph to represent these types of data.
As for the negative side effects associated with IFN, we discussed these in terms of therapy for Hepatitis C virus which required prolonged (up to 48 weeks) use. For an acute infection like Dengue, a substantially shorter time frame would be required. Thus, the clinical regimen of IFN would likely be ideal at this point to result in maximal inhibition of viral replication in the three cell lines, since antiviral activity is so variable between cell types.